# Linking Beekeepers’ and Farmers’ Preferences towards Pollination Services in Greek Kiwi Systems

**DOI:** 10.3390/ani13050806

**Published:** 2023-02-23

**Authors:** Elie Abou Nader, Georgios Kleftodimos, Leonidas Sotirios Kyrgiakos, Christina Kleisiari, Nicola Gallai, Salem Darwich, Tristan Berchoux, George Vlontzos, Hatem Belhouchette

**Affiliations:** 1CIHEAM-IAMM, UMR CEE-M, F-34093 Montpellier, France; 2CEE-M, University of Montpellier, CIHEAM-IAMM, CIRAD, INRAE, Institut Agro, F-34090 Montpellier, France; 3Department of Agriculture Crop Production and Rural Environment, University of Thessaly, 38446 Volos, Greece; 4LEREPS, ENSFEA, Université Fédérale Toulouse Midi-Pyrénées, F-31042 Toulouse, France; 5Faculty of Agriculture, Lebanese University, Beirut 99, Lebanon; 6CIHEAM-IAMM, UMR TETIS, F-34093 Montpellier, France; 7CIHEAM-IAMM, UMR ABSys, F-34093 Montpellier, France

**Keywords:** pollination services, ecosystem services, beekeepers, public policy, farmer’s decision making

## Abstract

**Simple Summary:**

Greek kiwi production systems suffer from a Pollination Services (PS) shortage due to the declining number of wild pollinators. This study assesses the barriers towards the implementation of a PS market in Greek kiwi production systems by conducting two separate quantitative surveys, one for beekeepers and one for kiwi producers. The field survey findings corroborate the existence of a strong basis for further collaboration between the two stakeholders, as both of them acknowledge the importance of PS. Moreover, the farmers’ willingness to pay and the beekeepers’ willingness to receive regarding the renting of their hives for PS were examined.

**Abstract:**

The kiwi is a highly insect-pollinated dependent crop and is the cornerstone of the Greek agricultural sector, rendering the country as the fourth biggest kiwi producer worldwide, with an expected increase in national production the following years. This extensive transformation of the Greek arable land to Kiwi monocultures in combination with a worldwide shortage of pollination services due to the wild pollinators’ decline raises questions for the provision of pollination services, and consequently, for the sustainability of the sector. In many countries, this shortage of pollination services has been addressed by the installation of pollination services markets, such as those in the USA and France. Therefore, this study tries to identify the barriers towards the implementation of a pollination services market in Greek kiwi production systems by conducting two separate quantitative surveys, one for beekeepers and one for kiwi producers. The findings showed a strong basis for further collaboration between the two stakeholders, as both of them acknowledge the importance of pollination services. Moreover, the farmers’ willingness to pay and the beekeepers’ willingness to receive of the beekeepers regarding the renting of their hives for pollination services were examined.

## 1. Introduction

Pollination Services (PS) are a vital ecosystem service for agricultural systems, as more than the 75% of the global crops destined for food consumption depend on them [1]. In general, pollination services are provided by a wide range of animals such as bees, butterflies, moths, flies, beetles, birds and mammals [2], however, bees, which are the predominant and most economically important group of pollinators, are facing a substantial population decline worldwide, and particular, in Europe [3,4,5]. This decline has been triggered due to several motives and phenomena attached to agricultural activity, such as intensified pesticide use, monoculture, the deterioration of natural habitats, etc. [3]. Consequently, the continuity of the above practices, in combination with the increasing agricultural land devoted to insect pollinator depended crops [5], may lead to a pollination crisis according to which crop yields starts to fall due to inadequate pollination services [6,7].

In this context, the European Union, through the Common Agricultural Policy (CAP), launched a wide range of policy measures in order to protect insect pollinators. These measures include a variety of costly Agri-Environmental Measures (AEM) [8], a regulation (EU No 485/2013) banning the use of three neonicotinoids in agricultural systems, which are responsible for bees’ decline and the Pollinators Initiative. This last measure aims to alter the decline of wild pollinators through a series of actions, focusing mostly on raising knowledge and awareness of the pollinators’ importance and addressing the main stressors, such as pesticides [9]. Hence, according to this initiative, every Member State has the opportunity to identify the main issues and design and implement frameworks for the protection of wild pollinators.

However, all of the aforementioned policy measures seem ineffective to protect pollinators and safeguard the provision of pollination services in arable crop farms in the long term [8,10]. For instance, the majority of the implemented AEMs proposed focus only on well-established bee species, such as bumblebees or honey bees and neglect wider pollinators species, such as solitary bees [11,12]. As a result, these measures may lead to the extinction of a variety of insect pollinators that are valuable for the functioning of ecosystems, and consequently, may negatively affect the pollination services provided by bumblebees and honey bees [12]. In addition, according to the report of the European Court of Auditors, the Pollinators Initiative is not a regulation, but a communication, and will/may consequently fail to mobilize the majority of the Member States to establish a legal framework for the provision of wild pollinators [10].

Therefore, the continuous decline of wild pollinators in combination with the ineffective policies are driving many agricultural systems to depend more and more on renting/buying managed pollinators in order to obtain sufficient pollination services [13,14]. Indeed, a market of pollination services, such as in the USA [15], has recently emerged in the south west of France in the Occitanie region [16,17]. In fact, the creation of pollination services markets may offer a solution to the farmers’ needs for pollination services, while at the same time, it can offer alternative sources of incomes for the beekeepers [18]. This is a traditional practice in the USA, especially for almond production [19]. However, in Europe, these markets are quite new, so there is a significant lack of information regarding prices, organization, transaction costs, contracting, etc. 

Thus, it is of paramount importance to identify the barriers to the creation and the functioning of these markets. For this purpose, it is necessary to understand the decision-making process of beekeepers and farmers. By understanding their perceptions regarding pollination services and their mode of actions, it may be easier to facilitate the collaboration of the two stakeholders, as well as to promote effective policy measures for the protection of wild pollinators.

In this study, we are examining the farmers’ and beekeepers’ perceptions towards pollination services in Greek kiwi production systems. In fact, Kiwi is a highly bee-pollination-dependent crop [1] and is one of the cornerstones of the Greek agricultural sector, rendering the country as the fourth biggest kiwi producer worldwide, with an expected increase in national production in the following five years [20]. This extensive transformation of the Greek arable land to Kiwi monocultures in combination with wild pollinator decline raise questions for the provision of pollination services, and consequently, for the sustainability of the sector. Thus, here we analyze the conditions for the implementation of pollination services markets in Greek kiwi production systems in the region of Kavala, determining the perceptions of the stakeholders.

In order to do so, we conducted two separate quantitative surveys, one for beekeepers and one for kiwi producers, in order to collectively explore (i) the possibility of collaboration between the stakeholders, (ii) the stakeholders’ perception of pollination services and (iii) the stakeholders’ decision making regarding pollination services.

The first section justifies the selection of the examined case study and presents a step-by-step analysis of the elaborated methodology. The second section presents the obtained results, while the third section discusses the main findings. The final section summarizes the main conclusions, presents the limitations and discusses the perspectives for future research.

## 2. Materials and Methods

In this section, we present: (i) the selection of the studied area; (ii) the data collection; (iii) the elaborated statistical analysis.

### 2.1. Study Area

As we mentioned above, kiwi is one of the most important and dynamic crops of Greek agriculture, with 250,140 tons being produced in 2019. The most important kiwi regions are Pieria, Kavala and Arta, producing the 31%, 15% and 26% of the national production, respectively. Moreover, in the regions of Arta and Kavala, an increasing number of new kiwi plantations have been observed the last decade in the place of oranges, peaches, apricots, etc. (https://www.statistics.gr/en/home/, accessed on 1 July 2021). Hence, in both regions, a significant increase in the number of kiwi trees is further anticipated in the near future. 

This rapid transformation of these two regions to kiwi monocultures in combination with the widespread pollinators’ decline in European landscapes [21] raise questions about the future provision of pollination services, and consequently, the sustainability of the sector. Between these two areas, the Kavala region is characterized as one of the most important beekeeping areas in Greece, with more than forty-six thousand registered beehives and a long tradition in beehive products. Moreover, the majority of the professional beekeepers are located in the island of Thasos, moving their hives long distances within the wider region of Eastern Macedonia and Thrace during the year in order to find available nectar resources [22]. Therefore, for all of the aforementioned reasons, the region of Kavala is a suitable case study, as it combines strong kiwi production and a significant presence of nomadic beekeepers, and consequently, it is possible that pollination services market may emerge in the near future. 

In general, the prefecture of Kavala is located in the wider region of Eastern Macedonia and Thrace in the north east of Greece. It is divided in three municipalities, the municipalities of Kavala, Nestos, and Pangaio, and the minor prefecture of the island of Thasos (https://www.pamth.gov.gr/index.php/en/ accessed on 1 July 2021). Despite the fact that kiwi plantations are widespread in this regional unit, the most intensive systems are located in the municipality of Nestos. This municipality has a particular geographical feature: it is located in a valley with numerous agricultural systems and a great variety of agricultural products [23]. Moreover, the River Nestos is traversing this municipality and forms an extensive Delta before it flows to the northern Aegean Sea, irrigating more than 40,000 ha of agricultural land. As a result, the primary economic section of the municipality of Nestos is agricultural activity, with a high rate of production of kiwi fruits, as shown by Sykianakis et al., 2019 [24]. Finally, located in the south of this municipality is the city of Keramoti, which is the main connection port with the island of Thasos and facilitates the movement of beekeepers throughout the year. In this selected case study, through our questionnaires, we identified two geographical areas where the kiwi producers are located (Figure 1). The area (A) is located in the western part of the municipality of Nestos, while the area (B) is on the east is within a close distance away from the Nestos river and a forestry area.

### 2.2. Data Collection

Data have been collected through face-to-face interviews by using two types of questionnaires each for kiwi producers (hereafter, ‘farmers’) and beekeepers. A small group of 5 kiwi producers and 6 beekeepers were interviewed to test and calibrate the questionnaire, but only minor changes were necessary to be applied. Both questionnaires had a similar format and were inspired and adapted by the study by Breeze et al. (2019) [25]. The first part of the questionnaire was the same for both groups, referring to the demographic characteristics of the respondent (age, gender and educational level), number of hectares or beehives, whether they were amateurs or professional (for beekeepers) and the location of agricultural activity. The following section, regarding farmers, assessed the pollination services deficit attitudes, the implemented strategies to ameliorate pollination services (if any) and the prices for these services. Similar questions were included in the beekeepers’ questionnaire about the reasons why they rent their beehives, the price and the crops they prefer or avoid. Moreover, a series of open questions have been used to gain further insights from each group about their beliefs regarding pollination service increases or its perceived limitations. Furthermore, willingness to pay (WTP) and willingness to receive (WTR) were examined to identify any contradiction between the two groups. Finally, the last part had a series of open questions in order to examine the farmers’ and beekeepers’ perceptions towards the adoption of policy incentives for the provision of pollination services in agricultural systems. Farmers were asked about what types of management strategies or measures they are willing to adopt in order to attract more beekeepers to rent their hives, as well as to identify practices for the provision of wild pollinators within their farmlands. The beekeepers were asked to name what types of policy incentives may encourage them to increase the number of hives in order to offer more pollination services to the farmers. All of the answers were recorded anonymously, without any storage of personal information. 

The survey was conducted between June and August 2021. Questionnaires were distributed to collaborations of local farmers’ cooperatives and beekeepers’ cooperatives. The names and basic structure are the following: i.EAS Kavala cooperative (Farmers’ cooperative) (250 members, 350 ha);ii.Cooperative Municipality of Nestos (Farmers’ cooperative) (60 members, 100 ha);iii.Nespar cooperative (Farmers’ cooperative) (51 members, 150 ha);iv.Apiculture cooperative of Kavala (Beekeepers’ cooperative) (230 members);v.Beekeeping Cooperative of Thasos (Beekeepers’ cooperative) (104 members).

Out of the 334 beekeepers in total in the region, 49 questionnaires were collected, representing 14.6% of the total, whereas for the kiwi producers from the Cooperation Municipality of Nestos, which has in total 100 ha of kiwi orchards, 5 members (8.3%) were interviewed, who cultivated, in total, 20 ha of kiwi (20%). In addition, 31 farmers (12.4%) were surveyed from the EAS Kavala cooperative, which had in total 250 members owning 169.8 ha of kiwi orchards or 48.5% of the total surface area of the cooperative. Furthermore, 11 kiwi producers from the Nespar cooperative (21.5%) managing 56.6 ha (37.7%). Overall, 13% of the total kiwi cultivated area in the valley of Kavala, which accounts for 1900 ha, was covered. In total 104 questionnaires were collected, but only 96 were accepted (47 kiwi producers and 49 beekeepers) after the data validation process. Additional information regarding the sample’s characteristics can be found in Table 1.

Descriptive statistics were extracted from all of the questions, and the most significant ones are visualized in the results section. Additionally, a Pollination Services indicator was created to assess the attitude of farmers towards pollination services. This indicator uses 4 questions from the survey related to (1) the perceived pollination services deficit, (2) the willingness to pay or willingness to increase the amount of money for pollination services, (3) the implementation of measures for encouraging wild pollinators (4) and planning to further support the existing wild pollinators by adopting alternative agricultural practices. Through the above-mentioned variables, the data type was collected, meaning that farmers either have a positive (1) or a negative (0) attitude towards PS. All of the variables were equally treated, without any use of weights since this indicator was used to describe a general overview of the farmers’ attitudes. Apart from the apparent grouping between beekeepers and farmers, a new dummy variable was introduced for separating the farmers regarding their proximity to the Nestos river. The farmers near the river or forest areas benefit from the pollination services of existing bees’ populations that are situated there due to the increased availability of nesting and foraging habitats, which is contrary to the farmers that are located further away, who do not benefit from these services. In other words, this dummy variable was added to highlight potential differences in the farmers’ attitude regarding pollination services between the two groups. Proximity to the Nestos river or forest areas was estimated within a 3 km radius, based on the effective flying distance of bees [26].

### 2.3. Statistical Analysis

In order to specify the existence of any differences between the medians of the formulated groups, a non-parametric approach has been selected by using a Mann–Whitney test [27], since the data were not normally distributed according to the Shapiro–Wilk test. The following comparisons have been made for farmers and beekeepers:(a).Number of beehives and provision of pollination services;(b).Farmers’ willingness to pay and beekeepers’ willingness to receive;(c).Farmers’ actual payments and beekeepers’ willingness to receive;(d).Current beekeepers’ and farmers’ prices for pollination services.

Moreover, Spearman’s rank order correlation coefficient (ρ) was used for assessing the relationships between the variables [28]. Any statistical correlation coefficient is described using integers ranging from −1 to 1; with −1 indicating a perfect negative relationship between two variables, while 1 indicates a perfect positive relationship. Spearman’s correlation was applied between the following variables for the two stakeholders:(a).Years of beekeepers’ experience and their perception of being professional or amateurs;(b).Years of beekeepers’ experience and the number of beehives;(c).Farmers’ management strategies to deal with pollination services deficit;(d).Farmers’ perception towards the impact of pollination services on kiwi yields.

All of the above statistical analyses were conducted with the use of R software, version 4.0.5 (base and stats package).

## 3. Results

In this section, we present our findings with regard to: (i) the socio-economic characteristics of the two stakeholders, as well as the current situation in the examined area, (ii) the beekeepers’ perception towards pollination services, (iii) the farmers’ perception of pollination services and collaboration with the beekeepers and (iv) the stakeholders’ perceptions towards public policy interventions.

### 3.1. Actual State of Pollination Services and Socio-Economic Characteristics of Beekeepers and Farmers 

The research sample consists of 49 beekeepers managing 7402 beehives, distributed in the prefecture of Kavala and the island of Thasos. On average, a beekeeper is 44 years old and owns 218 beehives. Eighty-five percent of the beekeepers interviewed were male, and fifteen percent were female. Sixty-eight percent of the beekeepers have a high school degree, and the rest hold a bachelor’s/master’s degree. Fifty-nine percent of the beekeepers considered themselves to be amateurs, while the remaining ones are professionals. On average, the beekeepers have 10 years of experience, and the majority (97%) move their hives to another location at least once a year (usually during summer). All of the beekeepers surveyed experienced an increase in demand for pollination services since 2018. Sixty-eight percent of the beekeeper sample rent their hives to local farmers, receiving in return an average of EUR 12 per hive. This price was based mostly on transportation and feed costs. In fact, kiwi orchards offer no nectar to the foraging bees and as a result, the beekeepers need to feed the bees themselves, and as a consequence, the farmers face higher maintenance costs. When they were asked if they were happy with this price, the beekeepers undoubtedly agreed that EUR 12 is not enough, demanding at least double that amount (EUR 24 per hive). Indeed, this finding was further validated by conducting a Mann–Whitney U test (W = 1100, *p* < 0.001) (Figure 2). Moreover, every beekeeper mentioned that this service takes place with no written contract. Indeed, 53% of these agreements happen orally, and 12% lend their hives to friends or to close relatives (Figure 1). However, the remaining 32% of them who do not provide pollination services had multiple reasons for this choice, such as producing organic honey (12%) or that they are not ready to take this step (6%). As anticipated, the beekeepers are not satisfied with this unofficial agreement, and 65% of them would like to negotiate on the rights and responsibilities of each party in terms of the number of hives per ha, the rental period, the cultivation practices (e.g., lower use of pesticides), the price per hive and the method of payment before proceeding to any exchange.

Regarding the kiwi producers, 47 farmers are located in the municipality of Nestos, managing 247 ha in total. One hundred and seventy ha were located in area (A), and seventy-six ha were located in area (B), closer to Nestos river and to the forestry area. The obtained results showed that, on average, a farmer is 46 years old and owns 5.2 ha of kiwi. Eight-seven percent of the farmers interviewed were male, and thirteen percent were female. Additionally, 70% have a high school degree, while the rest have a bachelor’s/master’s degree. All of the farmers affiliated with cooperatives are full time farmers. The data revealed that 66% of them agreed that they experience a pollination services deficit in their production systems. Fifty-five percent of them were unaware of the reason(s) leading to this deficit. The remaining 34% of the farmers stated that there was no pollination service deficit, and 15% justified their stance by mentioning the presence of beekeepers nearby or their close location to the Nestos river and forestry areas. In order to deal with the shortage of pollination services, 64% of the farmers adopt various strategies, such us renting beehives and buying managed bumblebees, while the remaining 36% do nothing.

The farmers take actions against the pollination services deficit, usually by renting hives (44%) for an average price of EUR 12 per hive or by buying bumblebees (34%) for an average price of EUR 64 per hive. It should be stated that private discussions with farmers have revealed that 15–20 hives are needed per hectare, a fact that is also confirmed by the literature [29]. Moreover, we observed two farmers buying “Flying Doctors” hives, a specific category of bumblebees for a price of EUR 200 per hive. In fact, this category of bumblebees is developed in specific hives with a dispenser which contains biopesticides, or previously collected kiwi pollen or even both, that sticks to a premium colony of bumblebees on their way out of the hive, and as a result, the bumblebees can deliver pollen, as well as microbial pesticides, to each flower they visit. Finally, a few farmers (22%) use both honeybee and bumblebee hives, while none of the abovementioned farmers have knowledge of solitary bees or other wild pollinators.

### 3.2. Beekeepers’ Perceptions of Pollination Services

As described in the Materials and Methods section, the beekeepers’ number of beehives and provision of pollination services were tested. Although non-significant results were obtained (*p* = 0.291), it is highlighted that there is no clear picture between which strategy should be adopted and whether to provide or not provide pollination services. 

Spearman’s correlation rank was used to identify the relationship between the variables. The beekeepers’ years of experience and their perception about being s professional or an amateur do not appear to have a significant correlation, meaning that there are several people with multiple years of experience that still consider themselves as amateurs. However, years of experience have a significant positive relationship (0.338) at significance level of α = 10% (0.051) with the number of self-owned beehives.

When it comes to the beekeepers’ preferences to crops, 18 crops were listed as ones which are preferable to use, with the most mentioned ones being pine, kiwi, thyme, oak, chestnut and almonds. Similarly, seven crops were listed as the best ones to avoid, with the most averted one being cotton, followed by olives and sunflowers. The main reason for this avoidance is the beekeepers’ perception of high use of pesticides in these crops. Surprisingly there was a significant overlap between the two groups, with three out of the seven crops that are commonly avoided also being among the eighteen most favored crops, which are kiwi, sunflowers and almonds (Figure 3).

In fact, the beekeepers’ main reasons for choosing a crop for hive placement are: (i) the production of higher quality honey, (ii) honey with a better commercial price, (iii) more honey production, (iv) the availability of nectar resources and (v) it being good for colony growth. In general, beekeepers prefer to place their hives close to wild flora, such as pine and thyme, as they perceive that these crops offer honey of better quality with a higher commercial price. However, when it comes to kiwi, 71% of beekeepers stated that the only reason for choosing this crop was the payments provided by farmers in return for their hives (Figure 4).

Advancing now to crops that are avoided by beekeepers, cotton obtained the most answers, with 56% of the responses, followed by olives and sunflowers, with 53%, and then kiwi, with 41%. The primary reason for crop avoidance is pesticide use, for example, cotton is widely unaccepted since it is associated with pesticide use (95% of answers) and poor-quality honey (5%). Similarly, sunflower was linked with pesticide danger (52%), and so were olives (28%). However, questions about sunflower and almonds received many controversial answers, as on the one hand, they offer valuable nectar resources for the bees, while on the other hand, they the farmers intensively use pesticides, which increases the bees’ mortality. Finally, 41% of them avoided kiwi orchard for two main reasons, the presence of other better pollen sources during this time of the year and the threat of pesticides.

### 3.3. Farmers Perception of Pollination Services and Collaboration with the Beekeepers

As already mentioned in the first part, 66% of the individuals surveyed claimed the presence of a pollination service shortage in their orchard, but when they were asked for the reasons behind this statement, 84% failed to respond, 10% had no idea what was behind this occurrence and only 6% associated this deficit with the lack of insect pollinators. The farmers who denied the presence of a pollination services deficit backed up their statement with more reasons than the farmers who thought there was a deficit did. Likewise, the majority of them (44%) gave no answer, 19% mentioned the presence of beekeepers nearby, another 19% justified their stance with their close location to the Nestos river and forestry areas, where there is beekeeping activity and high abundance of wild pollinators, 13% blamed the structure of the kiwi orchard, which contains many unproductive male plants rather than an insufficient number of pollinators, and the rest (6%) stated ignorance (Figure 5).

Indeed, regarding the farmers who responded that they do not experience a pollination service deficit, location seems to be a crucial factor. Hence, we conducted a Mann–Whitney test based on the PS Indicator (mentioned in Section 2) in order to examine the effect of location on the farmers’ perception for pollination services deficit (Table 2). With a significance of 99.5% (W = 125, *p* = 0.005), the results showed that there is a clear difference between the perceptions of kiwi producers situated close to Area (B) compared with those of the other group. This result was expected, since the farmers from Area (B) mentioned that a lot of beekeepers place their hives near the Nestos river, and consequently, they positively benefit from these hives. Moreover, the nearby forestry areas provide a large number of natural habitats, and as a result, wild pollinators are more likely to visit these kiwi systems. The PS indicator was compared to the demographic characteristics and farm size as well, but no statistically significant relationship arose.

Regarding the importance of pollination services, our results highlight that the majority of farmers (30%) believe that adequate levels of pollination services increase the kiwi yields by 20–50%, 23% perceive an increase of 10–20%, 28% perceive an increase of 0–10%, 13% believe that pollinators contribute more than 50% to the kiwi yields, and the rest stated that it is not important for production. Hence, the great majority of the farmers believe that pollination services are necessary for improving productivity of their crops (Table 2).

The farmers who acknowledged a pollination services deficit adopt several management strategies to address it. In fact, the majority of them (44%) rent beehives from local beekeepers, while others rent managed bumblebees (34%) or use a combination of the two bee species (22%).

In assessing the motives behind the utilization of honeybees or bumblebees, the farmers mostly answered that honeybees as pollinators are more effective (34%) than bumblebees are, despite them being recommended by the local agronomists (23%), while completely denying the benefits of wild pollinators. 

Moreover, in order to further examine the farmers’ strategy to deal with pollination services shortage and their perceived increase in kiwi productivity, a Spearman correlation was applied. Between those two variables, a non-parametric correlation with the highest degree of significance was found (*p* = 0.003). In fact, farmers invest in beehives to acquire a higher yield. In other words, pollination services are adapted not for environmental protection, but primarily for the economic development of local production systems. It should be noted that this could be a way to persuade the beekeepers to provide more beehives to local kiwi producers who will incorporate them as a key input in their kiwi production (Table 3).

In general, farmers are willing to increase their purchase of beehives from local beekeepers in order to maintain sufficient pollination services in their production systems. Indeed, 62% of the farmers revealed their willingness to start paying or even pay more than they are currently to receive more hives. For instance, the data show that half of them (52%) were willing to pay between EUR 0–20 per hive, another 38% were willing to pay EUR 20–50 per hive, and lastly only 7% were willing to pay EUR 50–100 per hive. Additionally, farmers have already embraced the importance of pollination services, as the majority of them (78%) have already reduced their current pesticide use in order to encourage beekeepers to supply more hives.

After noticing the willingness of farmers to pay or pay more for renting hives from local beekeepers, we conducted a Mann–Whitney test between the willingness of beekeepers to receive and of the farmers to pay EUR 24 per hive. This price has been detected through our questionnaires as the average price which will motivate the beekeepers to continue the provision of pollination services to kiwi production systems. Our findings highlighted no significant difference, and consequently, the value of EUR 24 per hive is commonly accepted by the two stakeholders (W = 155, *p* = 0.000) (Table 4).

### 3.4. Stakeholders Perceptions towards Public Policy Interventions

Regarding public policy interventions, there is no program focused on the pollination services in the greater region of Eastern Macedonia and Thrace [22]. Hence, in order to examine the possibility of implementing such a measure, we proposed a set of open questions focused on the stakeholders’ views towards their participation in public policy incentives, targeting the provision of pollination services. All of the respondents welcomed the implementation of new policy measure, such as Agri-Environmental Schemes (AES) or Payments for Ecosystem Services (PES). 

When it comes to beekeepers, a premium on the renting price per hive (18%) and the decrease in pesticide use by farmers (21%), including herbicides, or a combination of these two measures (15%) are key elements of any new public policy. Concerning farmers, 87% of them expressed their interest in participating in public policy incentives, such as AESs or PES, to increase the supply of pollination services. Moreover, they responded that they are willing to adopt alternative management strategies in order to increase the farmland biodiversity and benefit from wild pollinators by installing wildflower strips, nesting sites, etc. However, their main obstacles for the adoption of these measures are: (i) 38% labor availability (38%), (ii) a lack of necessary equipment (11%), (iii) a lack of experience (19%) and (iv) possible emerging costs (32%). In addition, they insisted that they are taking of all the necessary precautions regarding pesticides use, something that contradicts the beekeepers’ perceptions. In fact, when it comes to pesticides, farmers perceive only the use of insecticide, neglecting the negative impacts of the herbicides on the bee pollinators [30]. In contrast, the regional beekeepers are well aware of these impacts and made a special reference to these active ingredients.

## 4. Discussion

In this paper, we examine the beekeepers’ and kiwi producers’ perceptions towards the provision of pollination services. Our findings signify the existence of an informal market of pollination services in the region of Kavala, however, both stakeholders seem to be unsatisfied with their current collaboration. 

In fact, beekeepers usually select crops that offer high-quality honey with a higher commercial value, thus optimizing the production process. Moreover, they also move their hives once per year according to the need for available foraging resources for their colonies. Moreover, regarding crop avoidance, their perceived biggest threat is to place their hive in an area with crops with high pesticides use, such as cotton, sunflower or olives. However, we detected an overlap in the beekeepers’ decisions, as many crops, such sunflower or almonds, are also used and avoided by them. This overlap is based on the fact that these crops provide good sources of nectar, especially in periods when other floral resources are not available, suggesting that the beekeepers’ actions are mainly based on their own experience regarding pesticides use, rather than based on scientific findings. In addition, the majority of regional beekeepers choose to place their hives next to kiwi production areas only when they receive payment by local producers. However, providing PS to kiwi producers is rather limited as the payments are too low to cover their high traveling and maintenance costs. Consequently, they provide only a small number of hives and only when other pollen resources are not available. Furthermore, despite the fact that farmers demand a higher supply of pollination services, the beekeepers are not willing to provide them due to the high risk of pesticides. The above findings are in accordance with the study by Breeze et al. (2019), who that mentioned that coordination actions and higher prices are needed in order to attract a higher number of beekeepers to offer pollination services. 

Regarding the farmers’ perception, the majority of them acknowledge the importance of pollination services for kiwi production and declared that they experience a pollination services deficit. Moreover, a key element to their perceptions was the location of their production systems. Thus, they are willing to increase the provision of beehives from local beekeepers by increasing the existing prices. Indeed, the willingness of farmers to pay for pollination services is close to the willingness of the beekeepers to receive, and consequently, the basic conditions for the creation of a pollination services market are met. This finding is in accordance with the literature, which proposes that higher payments for pollination services may address the issues of pollination deficit [14,30]. Furthermore, this is the first case study in Europe where both stakeholders acknowledge the importance of pollination services, and their perceptions of the renting price are the same. Hence, the development of such a market may arise naturally through extensive dialogue between the two stakeholders.

However, the collaboration between different stakeholders is not given, as many socio-economic and socio-cultural issues may exist. Several studies suggest that the coordination of different actors, especially on payments for ecosystems services such as pollination, demands the cognitive and normative legitimation of the proposed actions. In other words, both stakeholders have to engage in social interactions in order to share and objectify their beliefs to establish new rules regarding what they can or cannot do in terms of practices and what they should or should not do in terms of moral values [26,31]. 

Towards that direction, the biggest problem to overcome is the use of pesticides. According to our findings, on the one hand, the farmers believe that they take all of the necessary measures to protect the bees, however, they do not believe that herbicides are toxic to them. On the other hand, the beekeepers do not trust the farmers and they do not want to offer a larger supply of beehives to kiwi systems. Consequently, a policy measure or a PES scheme should be implemented in order to facilitate the dialogue and mobilize the farmers to take actions for the protection of bees, while at the same time, it provides incentives to the beekeepers in order to reduce their production costs and support the declining apiculture industry [32,33]. According to the obtained results, both stakeholders are willing to be engaged in such measures, in the form of AESs or PESs, in order to increase the profitability of their sectors. 

Apart from the economic gains themselves, such a measure may have several environmental outcomes. Providing additional insights on this, the municipality of Nestos has a variety of different production systems, however, in recent years, there had been a tendency for new plantations of kiwi trees. A lack of public intervention will probably lead to a transition to a kiwi monoculture, which will reduce the available nectar resources. Consequently, honey production will decline, while the maintenance costs of the beehives will be significantly increased, as beekeepers will have to feed them more often [26]. Therefore, it is significant that public authorities should intervene and regulate this market in order to guarantee the existence of a diverse agricultural landscape with varied foraging resources. This statement is in accordance with Batáry et al. (2015), who support the implementation of AESs for supporting the provision of pollination services in agricultural landscapes [8].

Finally, even though a pollination service market may be beneficial for both farmers and beekeepers, it may also affect the wild pollinators negatively. In fact, the establishment of the market may conclude with the introduction of large numbers of honeybees in the landscape, which in turn, will compete with native wild pollinators for the available foraging habitats, and consequently, drive them to extinction [33,34,35,36]. Hence, the intervention of a public policy is important to regulate the number and the placement of beehives. Moreover, this policy may also motivate the farmers to maintain the natural habitats within their farmlands to provide suitable nesting sites for wild pollinators and profit from the wild pollination services. In fact, wild pollinators are more effective than honeybees are, while the presence of both bee species in the farmland may increase the yield outcomes, and consequently, the economic profit of the farmers [17,32].

Taking into consideration all of the above-mentioned results, we can state that while base conditions for the establishment of pollination services market exist in the region of Kavala, there is an increased need for public policy intervention. This may be the opportunity for the establishment of the first AES for pollinators in Greece, following the Pollinator Initiative (https://ec.europa.eu/environment/nature/conservation/species/pollinators/policy_en.htm accessed on 1 July 2021). The need for policy intervention is imminent, especially after the wildfires in 2021 on the island of Euboea, which may drive many nomadic beekeepers in the region of Kavala to search for available nesting sites or to offer pollination services to gain an extra income.

## 5. Conclusions

In this study, we examined farmers’ and beekeepers’ perceptions towards the implementation of pollination services markets in Greek kiwi production systems in the region of Kavala. Hence, two separate quantitative surveys were conducted, one for beekeepers and one for kiwi producers, in order to collectively explore (i) the possibility of collaboration between the stakeholders, (ii) the stakeholders’ perception of pollination services and (iii) the stakeholders’ decision making regarding pollination services. 

Our findings indicate that both stakeholders acknowledge the importance of pollination services for kiwi production and that there is indeed an unofficial market for PS in the region. However, the current situation is unpleasant for both stakeholders, as farmers are demanding a higher supply of beehives, while beekeepers are avoiding kiwi systems due to pesticide use and the absence of nectar resources. Moreover, we found that the only reason to rent beehives out to kiwi systems is the received payment. Towards that direction, we presented that farmers are willing to increase their payments in order to ensure that beekeepers place their hives within their production systems. Indeed, the WTP of the farmers overlaps with the WTR of the beekeepers, and consequently, the main principals for the establishment of a sustainable PS markets meet. 

However, the main obstacle for the establishment of this market is the use of pesticides. Policy implementations could play a major role by initiating the dialogue between the two stakeholders and support the establishment of a PS market, which could positively contribute towards the economic and environmental viability of the regional production systems. Our analyses indicate that both stakeholders are willing to participate in such a policy measure. 

Nevertheless, it should be noted that our findings depend only on the stakeholders’ perception of pollination services, and there is no study regarding the actual provision of pollination services from wild pollinators or honeybees in the landscape. Additionally, there is no information regarding the public sector officials’ willingness to formulate and apply the supportive measures that already exist in other EU countries.

## Figures and Tables

**Figure 1 animals-13-00806-f001:**
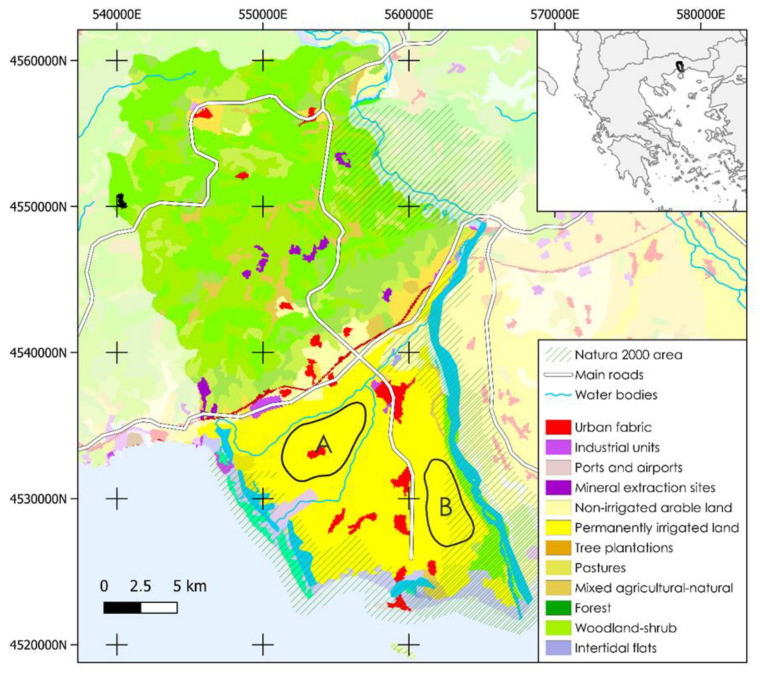
Location of the two study areas in the municipality of Nestos. Area A is located near urban and road infrastructures, while area B is adjacent to a large water body and woodland, designated as a Natura 2000 protected area. Data sources: Corine Land Cover (2018), EEA (2022) and GADM (2022).

**Figure 2 animals-13-00806-f002:**
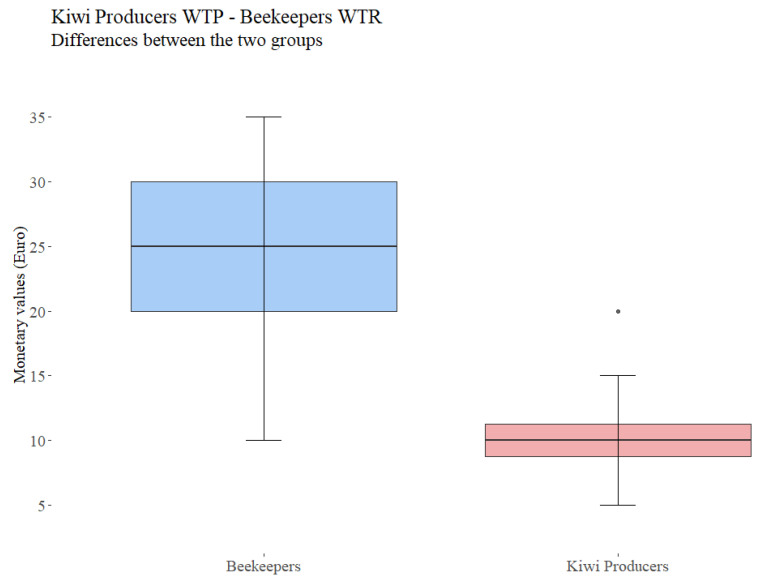
Mann–Whitney U test for current price and beekeeper’s willingness to receive in exchange for hives supply.

**Figure 3 animals-13-00806-f003:**
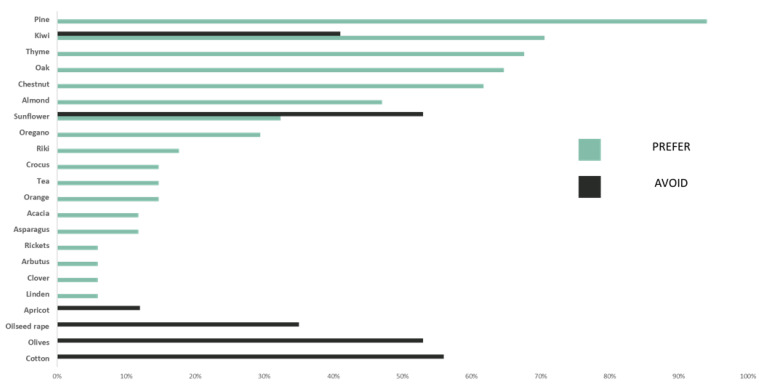
Summary of the main crops used (blue) and avoided (black) by beekeepers.

**Figure 4 animals-13-00806-f004:**
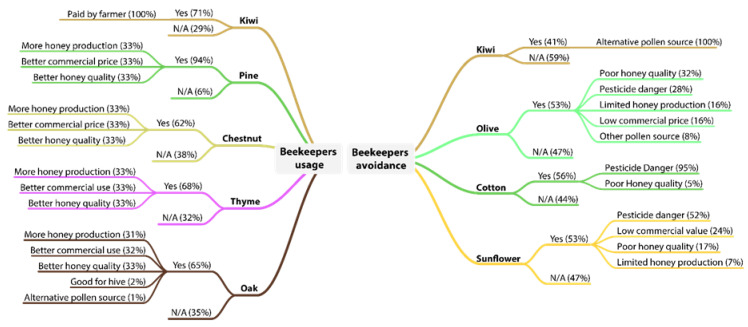
Beekeepers’ usage and avoidance of different trees and crops.

**Figure 5 animals-13-00806-f005:**
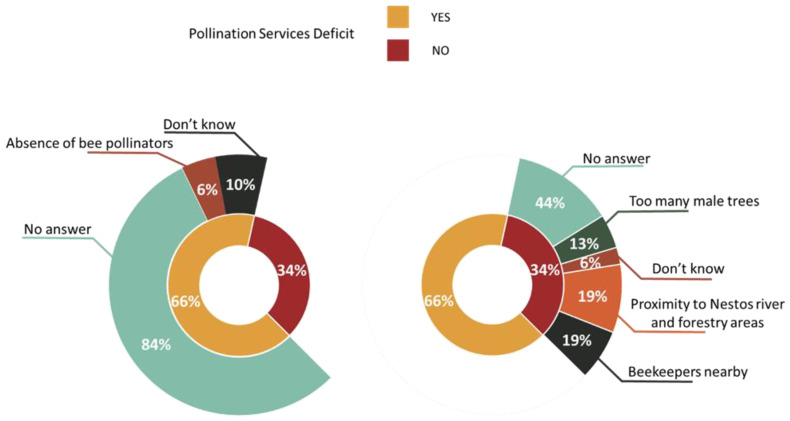
Farmers’ perception towards pollination services deficit.

**Table 1 animals-13-00806-t001:** Samples’ characteristics compared to agricultural/beekeepers cooperatives structure.

Samples’ Structure Compared to Agricultural/Beekeepers Cooperatives					
	Agricultural Cooperatives’ Structure	Sample Structure		Beekeepers Cooperatives’ Structure	Sample Structure
Farm size (ha)	N of farmers (%)		N of beehives	N of beekeepers (%)	
0–3	97 (27)	12 (26)	<200	37 (11)	7 (15)
3–6	191 (53)	26 (56)	200–400	224 (67)	35 (71)
6–9	69 (19)	8 (16)	400–600	53 (16)	6 (12)
>9	4 (1)	1 (2)	>600	20 (6)	1 (2)
Total	361	47	Total	334	49
Gender			Gender		
Male	321 (89)	41 (87)		70 (21)	7 (15)
Female	40 (11)	6 (13)		264 (79)	42 (85)
Total	361	47	Total	334	49

**Table 2 animals-13-00806-t002:** Mann–Whitney U test across farmer’s current location and the PS indicator.

PS Indicator Scores Across Locations		
Kavala (N = 32)	Scores	Nestos River (N = 15)
4	4	0
10	3	0
13	2	10
4	1	3
1	0	2

**Table 3 animals-13-00806-t003:** Spearman’s correlation between present strategies and perceived increase in kiwi production.

Spearman’s Correlation between Present Strategies and Perceived Increase in Kiwi Production	Dealing with Shortage	Values
Perception of increased production with the use of PS	Comparison Coefficient	0.425
	Sig. (2-tailed)	0.003
	N	47

**Table 4 animals-13-00806-t004:** Mann–Whitney U test across farmers and beekeepers’ current price in exchange for hives for pollination services.

Kiwi Producers WTP vs. Beekeepers WTR		
Kiwi Producers (N = 16)	Price (EUR)	Beekeepers (N = 34)
2	20–22	1
0	18–20	0
0	16–18	0
2	14–16	4
1	12–14	4
8	10–12	6
0	8–10	0
1	6–8	0
2	4–6	2
0	2–4	0
0	0–2	0

## Data Availability

Not applicable.

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
