# Peer review of "Linking Beekeepers’ and Farmers’ Preferences towards Pollination Services in Greek Kiwi Systems"

_animals, 2023, doi:10.3390/ani13050806_

Round 1
Reviewer 1 Report
It's a very interesting and modern subject and the paper is very well structured.
My suggestions:
- line 158: I suggest "demographic characteristics" instead of "social characteristics"
lines 185-194: Please provide more information on the methodology applied: how was the sample selected? Were there any criteria, such as age, education, farm size etc?
lines 253-259: Do you know how many farmers are full time farmers and how many have farming as a secondary occupation? It's an important information.
lines 289-290: How many hives are needed per hectare? It's an important missing information.
Has the relationship of the indicator with the demographic characteristics of the sample been checked? That is, if age or educational level affects the views of the sample?
Author Response
My suggestions:
Comment 1: line 158: I suggest "demographic characteristics" instead of "social characteristics"
ANSWER: Thank you very much for your comment. It has been modified in line 184.
Comment 2: lines 185-194: Please provide more information on the methodology applied: how was the sample selected? Were there any criteria, such as age, education, farm size etc?
ANSWER: Taking into consideration the structural and social characteristics of agricultural holdings participating in the cooperatives being studied in this survey, we selected applying the stratified sampling methodological approach. The representative sample which is described and analysed in this study. Additional evidence about those characteristics have been added into the text as it appears in Table 1.
Lines 187-189: “Additional information regarding the stratification of the sample can be found in Table 1.”
Comment 3: lines 253-259: Do you know how many farmers are full time farmers and how many have farming as a secondary occupation? It's an important information.
ANSWER: Lines 286-287: “All farmers participating in the specific cooperatives are considered as full time ones.”
Comment 4: lines 289-290: How many hives are needed per hectare? It's an important missing information.
ANSWER: Both farmers and literature indicate that 15-20 hives are needed for the provision of sufficient pollination services in the kiwi cultivation. Lines 320-322
Comment 5: Has the relationship of the indicator with the demographic characteristics of the sample been checked? That is, if age or educational level affects the views of the sample?
ANSWER: Thank you a lot for your comment. The demographic characteristics as well as the size of the farm have been further analyzed to highlight any potential relationship. Unfortunately, only the location factor is considered significant. Your comment was embodied above Table 2.
Linear Regrassion Model Coefficients - Index |
|||||||||
Predictor |
Estimate |
SE |
t |
p |
|||||
Intercept ᵃ |
1.86514 |
0.95338 |
1.9563 |
0.058 |
|||||
Farm Size |
-0.00150 |
0.00484 |
-0.3095 |
0.759 |
|||||
Age |
-0.00257 |
0.01300 |
-0.1978 |
0.844 |
|||||
Education: |
|
|
|
|
|||||
1 – 0 |
0.73744 |
0.69519 |
1.0608 |
0.295 |
|||||
2 – 0 |
0.80229 |
0.65889 |
1.2176 |
0.231 |
|||||
3 – 0 |
0.80114 |
0.66140 |
1.2113 |
0.233 |

Reviewer 2 Report
I thank the authors and editors for giving me the opportunity to review this manuscript which presents the results of two surveys of beekeepers and kiwi growers in a region in Northern Greece. I learned a lot about pollination practices in kiwi in the region and think that the facts reported are of great interested to all researchers working on pollination in agriculture.
That said, I found the manuscript to be quite poorly written, to the point that the writing does a significant disfavour to the content and limits the likelihood of the study reaching its potential readership. The introduction and discussion are probably the worst passages and need to be rewritten if not by a professional English writer, at least by a native speaker. The style is also very imprecise at times, more akin to student prose than to academic work. An example of what I mean can be found on line 425:
"However, here we have to mention that the 47% of the despondence did not gave an answer in this question about the proposed measures, nevertheless, they are positive towards their participation."
The organisation of the presentation of results also needs to be improved for clarity as it gives the impression of jumping around topics randomly.
The statistical works seems to be well done and appropriate but sometimes the basic findings need to be presented first and more simply instead of being drowned by more complex tests. Some figures are excellent (4 and 5).
On the content, I find the facts reported here to be of great interested and worthy of publication if better presented.
One point I am not sure I understand is the value of pollination fees. It seems to be that $12 is the prevailing fee among surveyed beekeepers and growers. Of course, beekeepers will always say more would be better, but that is not useful information as it is most always true that sellers would like a higher price. Now, on figure 2, it seems that the ranges of WTP and WTA do overlap in the $12 area, which is consistent with the observed rate. However in the text, the authors talk about a $24 pollination fee and seem to imply that this would be acceptable to growers? This is discussed in the paragraph starting on line 400. I can not make sense of this point of argumentation.
More generally, I find that transactions between beekeepers and growers are poorly discussed and analysed throughout the manuscript. The authors seem to support the idea that both beekeepers and growers would be willing to transact at $24 per hive but for some reason are not. They then give some handwaving at the need for policy intervention of some sort, including maybe subsidies? Why would beekeepers and growers not transact if they both benefit and are willing to? This does not make sense to me and I am unable to tell if it is a flaw in the reasoning or a lack of clarity in its transcription into words.
The economics of pollination transactions has been well described by the small literature starting by Cheung 1973 and that includes the Rucker Thurman and Burgett (2012) used by the authors (ref number 19). I suggest that more of that small literature be drawn from in order to cast the discussion of pollination transactions, in terms of transaction costs, risk sharing (including pesticide risk) etc... (see for instance Goodrich 2019 or Champetier 2021 for a review).
Overall, I see a lot of potential in the content of the manuscript but hope that the authors will find the resources to rewrite at least the introduction, discussion and conclusion, and to improve the overall clarity and accuracy of its presentation. As it stands, the form of the manuscript does not, in my opinion, meet the standards of academic publication.
Reference suggested:
Cheung, S. N. (1973). The fable of the bees: an economic investigation. The Journal of Law and Economics, 16(1), 11-33.
Goodrich, B. K. (2019). Contracting for Pollination Services. Choices, 34(4), 1-13.
Champetier, A. (2021). Environmental Economics of Pollination. In Oxford Research Encyclopedia of Environmental Science.
Author Response
I thank the authors and editors for giving me the opportunity to review this manuscript which presents the results of two surveys of beekeepers and kiwi growers in a region in Northern Greece. I learned a lot about pollination practices in kiwi in the region and think that the facts reported are of great interested to all researchers working on pollination in agriculture.
That said, I found the manuscript to be quite poorly written, to the point that the writing does a significant disfavour to the content and limits the likelihood of the study reaching its potential readership. The introduction and discussion are probably the worst passages and need to be rewritten if not by a professional English writer, at least by a native speaker. The style is also very imprecise at times, more akin to student prose than to academic work. An example of what I mean can be found on line 425:
ANSWER: Thank you very much for your comment. The whole text has been proofread by the services of our organization.
"However, here we have to mention that the 47% of the despondence did not gave an answer in this question about the proposed measures, nevertheless, they are positive towards their participation."
The organisation of the presentation of results also needs to be improved for clarity as it gives the impression of jumping around topics randomly.
ANSWER: Thank you for your suggestions. The text has been modified and several additional information has been added to both methods and results sections.
The statistical works seems to be well done and appropriate but sometimes the basic findings need to be presented first and more simply instead of being drowned by more complex tests. Some figures are excellent (4 and 5).
On the content, I find the facts reported here to be of great interested and worthy of publication if better presented.
One point I am not sure I understand is the value of pollination fees. It seems to be that $12 is the prevailing fee among surveyed beekeepers and growers. Of course, beekeepers will always say more would be better, but that is not useful information as it is most always true that sellers would like a higher price. Now, on figure 2, it seems that the ranges of WTP and WTA do overlap in the $12 area, which is consistent with the observed rate. However in the text, the authors talk about a $24 pollination fee and seem to imply that this would be acceptable to growers? This is discussed in the paragraph starting on line 400. I can not make sense of this point of argumentation.
ANSWER: Thank you for your question. In fact, at the price of €12 the provision of beehives is rather limited and does not cover the pollination services need. Under the price of €24 it is much more possible to attract higher number of beekeepers to offer pollination services. In any case, here we analyze the WTP of the farmers. Despite the existing prices, the analyses showed that the farmers are willing to pay more. For better understanding the discussion part has been modified.
More generally, I find that transactions between beekeepers and growers are poorly discussed and analysed throughout the manuscript. The authors seem to support the idea that both beekeepers and growers would be willing to transact at $24 per hive but for some reason are not. They then give some handwaving at the need for policy intervention of some sort, including maybe subsidies? Why would beekeepers and growers not transact if they both benefit and are willing to? This does not make sense to me and I am unable to tell if it is a flaw in the reasoning or a lack of clarity in its transcription into words.
ANSWER: Thank you for your question. The two stakeholders are not collaborating due to the perception of pesticides use. In stakeholders’ decision making there several determinants. Even if the cost-benefit determinant is respected, the perception for the pesticides use remains. This means that the farmers are going to keep use them in order to reduce the yield risk, while beekeepers are going to avoid them in order to minimize bees’ exposure. That’s why we need the public policy intervention.
The economics of pollination transactions has been well described by the small literature starting by Cheung 1973 and that includes the Rucker Thurman and Burgett (2012) used by the authors (ref number 19). I suggest that more of that small literature be drawn from in order to cast the discussion of pollination transactions, in terms of transaction costs, risk sharing (including pesticide risk) etc... (see for instance Goodrich 2019 or Champetier 2021 for a review).
ANSWER: Thank you for this comment. The introduction has been rewritten with respect in the proposed literature.
Overall, I see a lot of potential in the content of the manuscript but hope that the authors will find the resources to rewrite at least the introduction, discussion and conclusion, and to improve the overall clarity and accuracy of its presentation. As it stands, the form of the manuscript does not, in my opinion, meet the standards of academic publication.
ANSWER: Thank you for your suggestion. The introduction, discussion and conclusion have been modified. Moreover, the whole text has been proofread by the services of our organization.

Round 2
Reviewer 2 Report
The revisions on this manuscript do improve the ease of read. The summary and introduction paragraphs improve the presentation of the topic somewhat.
However, the revisions are in my opinion too superficial and too hastily executed. Aside from the three paragraphs, changes are mostly very superficial and do not address the concerns of organisation and clarity voiced in my initial review. Mistakes are still present, such as on line 621 where a "their" seems to have been lost, or in the first line of Table 1 which says "Stratification of the sample" which not accurate since the sample was not stratified on by design. Another example is the "will/may" on line 71. Yet another one is that line 617 and line 627 repeat the same comparison with the Breeze study.
Perhaps this is an issue of version and I am not looking at the final version of the resubmission but as I have read it, I don't find the manuscript to be ready for publication yet as the revisions have been insufficient so far, and even have added problems in some instances.
Still, I think the topic is important and the survey results of potential interests to reader.
Author Response
We thank the reviewer for his comments and his commitment to our work. According to his recommendations, we revised the results part in order to better present our findings. Moreover, following his suggestions, the discussion, and the conclusion parts have been rewritten in order to increase the clarity of our findings in relation to the existing literature.
More specifically:
- Regarding table 1: Table 1 was embodied, according to the comment of Reviewer 1 in the first round of revisions, in order to provide additional information for the samples' structures compared to the cooperatives' structure. We hope that the applied changes will be in accordance with your statement.
- The 2 paragraphs citing the work of Breeze and al. in the discussion part have been rewritten for better clarity.